# The Late Asthmatic Reaction Is in Part Independent from the Early Asthmatic Reactions

**DOI:** 10.3390/ijms26052088

**Published:** 2025-02-27

**Authors:** Stefan Zielen, Oguzhan Alemdar, Andreas Wimmers, Lucia Gronau, Ruth Duecker, Martin Hutter, Jordis Trischler, Jan G. de Monchy, Ralf Schubert

**Affiliations:** 1Department of Pediatrics, Division of Pneumology, Allergology, Infectious Diseases and Gastroenterology, Goethe University Frankfurt, 60590 Frankfurt am Main, Germany; oguzhan.alemdar@web.de (O.A.); a.wimmers@medaimun.de (A.W.); gronau@med.uni-frankfurt.de (L.G.); duecker@med.uni-frankfurt.de (R.D.); jordis.trischler@unimedizin-ffm.de (J.T.); r.schubert@med.uni-frankfurt.de (R.S.); 2Respiratory Research Institute, Medaimun GmbH, 60596 Frankfurt am Main, Germany; 3DC Klinieken Lairesse (Amsterdam), Valeriusplein 11, 1075 BG Amsterdam, The Netherlands; jan_de_monchy@hotmail.com

**Keywords:** early asthmatic and late asthmatic reaction, epigenetic regulation, microRNAs, bronchial allergen challenge, house dust allergy

## Abstract

House dust mites (HDM) are the world’s most important cause of allergic asthma. It is unclear why some patients with HDM allergy develop an early asthmatic reaction (EAR) only, whereas others react with a dual asthmatic reaction—EAR plus late asthmatic reaction (LAR). In patients with LAR, the symptoms and bronchial inflammation are more severe, and the current knowledge suggests that the EAR always precedes the LAR. The aim of the present study was to investigate whether a LAR can occur separately even without a significant EAR. In a pilot study of 20 patients with asthma and HDM allergy, a bronchial allergen challenge (BAC) was performed on three separate occasions with a tapered allergen dose. Before and 24 h later, exhaled NO (eNO), eosinophils and miRNAs were measured as markers of bronchial inflammation. Compared to BAC1, at BAC2 there was a significant decrease in the EAR from mean 39.25 ± 13.37% to mean 33.55 ± 5.25% (*p* < 0.01), whereas the LAR remained unchanged: mean 28.10 ± 10.95% to mean 30.31 ± 7.77% (n.s.). At BAC3, both the EAR and the LAR were significantly attenuated compared to the first and second BAC. In 3 (15%) patients, even the tapered allergen dose induced a dual asthmatic reaction. In 10 (50%) patients, the allergen dose was too low to trigger a significant EAR and LAR. In 7 (35%) patients, there was no EAR, but a significant LAR (mean max fall FEV1 20.5 + 4.7%) recorded. Significant correlations (*p* < 0.05) were found between distinct miRNAs (miR-15a-5p, miR-15b-5p and miR-374a-p5), eNO, and the decline in lung function and the presence of a LAR (*p* < 0.01). We can demonstrate that a LAR is induced in some patients without an EAR to low allergen exposure. This leads to a strong inflammatory reaction with an increase in eNO and a decrease in FEV1 and distinct miRNAs. Accordingly, these individuals are at greater risk of asthmatic symptoms and remodeling with loss of lung function than patients who do not have a LAR.

## 1. Introduction

Allergic asthma is the most common inflammatory airway disease in children and adults [1,2,3]. In allergic asthma, exposure to bronchial allergens leads to inflammatory and physiological manifestations of asthma, including airway obstruction, eosinophilic inflammation, and an increase in bronchial hyperreactivity (BHR) [4,5,6,7,8].

Inhalation of allergens in allergic asthmatics triggers an early asthmatic reaction (EAR) within minutes, which resolves within 1–3 h [7,8,9]. In addition, approximately 50–80% of asthmatics experience a prolonged late asthmatic reaction (LAR), which peaks within 3–9 h and resolves within 24 h [8,10,11]. The cellular mechanisms of EAR and LAR have been extensively studied: the EAR is triggered by the rapid release of histamine and lipid mediators such as leukotrienes, prostaglandins and the cytokines IL-4, IL-5 and IL-13 via the activation of local mast cells [7,12,13]. Bronchoconstriction, hypersecretion and swelling of the bronchial mucosa occur within minutes [7,8,14,15,16]. In contrast, the LAR leads to acute and chronic mucosal inflammation mediated by T lymphocytes, Treg cells and eosinophils [16,17,18]. This leads to further recruitment of inflammatory cells, especially TH-2 cells and eosinophil granulocytes, but also basophil and neutrophil granulocytes and macrophages [13,19,20]. Systemic and bronchial eosinophilic inflammation can be detected 24 h after bronchial allergen exposure [8,9,15]. The induced eosinophilic inflammation correlates with an increase in exhaled eNO [9,21,22]. The degree of BHR is closely related to the inflammatory response of the LAR and increases over several days [23,24,25]. This shows that in patients with allergic asthma, allergen exposure not only causes acute bronchospasms, but also triggers a longer-lasting cellular inflammatory response that can lead to an increased BHR [9,24,25,26]. In addition, we have recently shown that distinct mRNAs like miR-15a-5p, miR-15b-5p and miR374a-5p were significantly dysregulated in patients with an LAR [27].

Accordingly, recurrent or persistent allergen exposure leads to the development of chronic asthmatic inflammation with remodeling of the airways [28,29,30,31]. The important role for chronic house dust mite (HDM) exposure in children with allergic asthma was demonstrated by several studies [32,33]. The German MAS study was able to demonstrate that 90% of children with early childhood asthma/“viral wheeze” without allergy no longer had any symptoms at school age, whereas sensitization to HDM led to persistent asthma with a loss of lung function at school age [34]. In a study conducted by our working group, Donath et al. [35] were able to show that a high-grade BHR with simultaneous sensitization to HDM appears to be an independent risk factor for persistent asthma. Gender and asthma therapy had no influence on the development of BHR [35].

However, it is not exactly clear why some patients with HDM allergy develop only an EAR, while others respond with a dual asthmatic reaction—EAR plus LAR to a bronchial allergen challenge (BAC). If two distinct phenotypes of HDM allergy are present, this may have important clinical and public health implications. The phenotype with EAR only may be less likely to develop chronic asthma, whereas the LAR phenotype carries a high risk of progress to chronic asthma with loss of lung function. Accordingly, this pilot study investigated the question of whether a LAR may also occur in individuals without a significant EAR. If this phenomenon (LAR without significant EAR) can be detected more frequently, this would have clinical implications, since it is currently assumed that a LAR is always preceded by the EAR [7,8,15].

## 2. Results

### 2.1. Bronchial Allergen Challenge

As shown in Figure 1, the allergen dose was lower during BAC2 and BAC3 as compared to BAC1 (BAC1 median 30 AU; BAC2 median 19.06 AU and BAC3 19.06 AU). As expected, there was a significant decrease in the EAR from a mean 39.25 + 13.37% to a mean 33.55 + 5.25% (*p* < 0.01), whereas the LAR surprisingly remained unchanged: mean 28.10 + 10.95% to a mean 30.31 + 7.77% (n.s.). Only at BAC3 were both the EAR (8.29 + 3.47) and the LAR (12.51 + 7.07) significantly weaker than during the first two BACs.

### 2.2. Inflammatory Response (eNO and Eosinophils) After Reduced Allergen Doses

The eNO before BAC 1 was median 22 (5–81) ppb; in BAC2, median 41 (10–109) ppb and in BAC3, median 26 (8–108) ppb. Twenty-four hours after the BAP the eNO increased significantly in all BAPs: BAP1 median 58 (34–187) ppb, *p* < 0.001; in BAP2, median 77 (33–191) ppb, *p* < 0.001 and in BAP 3, median 46 (9–141) ppb, *p* < 0.001 (Figure 2a).

The eosinophils count before BAC1 was median 0.2 (0.05–54) nL; before BAC2 median 0.275 (0.14–1.13) nL and at BAP 3, median 0.185 (0.07–0.93) nL. Twenty-four hours after BAC, eosinophils increased significantly after all BACs: BAC1 median 0.425 (0.14–0.88) nL, *p* < 0.001; at BAC2, median 0.59 (0.19–1.58) nL *p* < 0.001, and at BAC3, median 0.31 (0.1–0.85) nL, *p* < 0.01 (Figure 2b).

### 2.3. Individual Reaction Patterns After the 3rd BAC

The following results were obtained for the individual patients during the BAC3:

In 3 (15%) of the patients, the allergen dose was too high and, as in the preliminary study, an EAR and LAR occurred. These patients were not analyzed further during the study.

In 10 (50%) of the patients (group 1), the allergen dose was too low to trigger an EAR (max fall FEV1 mean 6.8 + 1.9%) or LAR (max fall FEV1 mean 7.9 + 3.4%).

In 7 (35%) patients (group 2), there was no EAR (max fall FEV1 mean 9.65 + 2.68%) but a LAR (mean max fall FEV1 20.5 + 4.7%). The clinical characteristics of these patients are shown in Table 1. In addition, Table 2 shows that the bronchial inflammation was significantly more severe in patients with isolated LAR. Due to the small group size, significance was only found in the parameter’s delta eosinophilia and delta FEV1 drop.

### 2.4. MiRNAs

The comparison of the miRNAs, miR-15a-5p, miR-15b-5p and miR-374a-5p at time points t0, t7 and t24 demonstrated a significant downregulation of the expression rates (2^−ΔΔCT^) of all of the three miRNAs at time point t7, with a significant increase back to the initial level after 24 h (Figure 3). The miR-15a-5p expression before BAC (t0) was 0.98 (0.21–5.94), at t7 0.25 (0.2–0.86), *p* < 0.001 and at t24 0.67 (0.06–5.14), *p* < 0.01. Expression of miR-15b-5p before BAC (t0) was 0.97 (0.37–3.41), at t7 0.45 (0.2–1.34), *p* < 0.001 and at t24 0.57 (0.27–1.29), *p* < 0.01. The miR-374a-5p expression before BAC (t0) was 0.94 (0.22–5.58), at t7 0.29 (0.02–1.2), *p* < 0.01 and at t24 2.48 (0.2–8.05), *p* < 0.0001.

### 2.5. Correlation of microRNAs and eNO, Eosinophilia and Max Fall

In the next step, correlation analyses were carried out to investigate the relationship between the miRNAs miR-15a-5p, miR-15b-5p and miR-374a-5p in the blood with the inflammatory parameters eNO, eosinophils and lung function parameters (max fall during LAR) 24 h after BAC (Figure 4). Significant, negative correlations were found for all three miRNAs with eNO (miR-15a-5p: r = −0.59, *p* < 0.001; miR-15b-5p: r = −0.428, *p* < 0.05; miR-374a-5p: r = −0.482, *p* < 0.01). Only miR-15a-5p correlated with blood eosinophils (r = −0.361, *p* < 0.05), while both miR-15a-5p, miR- and miR-374a-5p showed a significant correlation with the max fall after LAR, and 15b-5p showed a trend towards a correlation (miR-15a-5p: r = −0.447, *p* < 0.01; miR-15b-5p: r = −0.308, *p* = 0.076; miR-374a-5p: r = −0.407, *p* < 0.05).

## 3. Discussion

In patients with mite allergic asthma, the inhalation of HDM induces an EAR, and in around 50–80%, an EAR and LAR [6,7,8]. In patients with a dual response, EAR and LAR, the symptoms and bronchial inflammation are much more severe after HDM exposure [11,15,17,18]. Late responses occurring in the absence of a preceding early response have been observed following occupational-type challenge tests for many years. However, recent information suggests that these isolated late responses may also occur frequently following exposure to common allergens at low concentrations, insufficient to provoke an immediate asthmatic [7,8,15]. However, if indeed this phenomenon (LAR without significant EAR) can be detected more frequently, this would have major clinical implications, since it is currently assumed that the EAR is in some way protective against further allergen exposure. To evaluate this phenomenon in more detail, the individual allergen dose of the BAC was gradually reduced. As expected, the allergen reduction of the BAC resulted in a significant decrease of the EAR, but surprisingly the LAR remained unchanged. As expected, a further reduction in the allergen dose resulted in no EAR in 50% of the patients. In contrast, 7 (35%) patients who did not experience an EAR did show a significant LAR with high eNO and drop in lung function after 24 h. Thus, we were able to demonstrate that a LAR can also occur without an initial EAR in a significant portion of HDM allergic patients.

Next, we carefully searched the literature and found some reports that a LAR can occur in isolated cases without an EAR [36,37,38]. A detailed review of all 356 publications on the keywords “late asthmatic reactions and allergen challenge” revealed a publication by Ihre et al. [38] from 1988 that largely agrees with our results. The authors investigated the occurrence of EAR and LAR after exposure to different allergen doses in 13 patients with mild or moderate symptoms of allergic asthma. Three different reaction patterns were observed: six patients (46%) showed an isolated LAR to relatively low allergen doses. In four patients (30.7%), an EAR plus LAR followed, and in three patients only an EAR occurred [38].

Based on our results, one-third of patients with asthma and HDM allergy experienced clinical symptoms 6–9 h after low allergen exposure and a flare-up of eosinophilic inflammation. The dogma that an EAR is usually perceived by those affected as a warning symptom and may only be followed by a LAR over time is challenged by our findings and the results of Ihre et al. [38]. Moreover, a silent trigger below the threshold of an EAR is barely noticeable and avoidable for the affected person after low allergen exposure. This phenotype constellation, where in some patients with allergic asthma a LAR is triggered even with low allergen exposure, certainly may contribute to the chronicity and remodeling of the airways with loss of lung function [29,30,31].

The clinical importance of the LAR is further demonstrated by increased TH-2 and Th17 cell cytokine production [26,39]. Th17 cells and eosinophils produce cytokines that promote airway inflammation and remodeling in allergic asthma [40,41]. This raises the question of whether patients with HDM and isolated EAR may suffer less from allergy and asthma than patients with EAR and LAR. Several investigators have investigated this association in patients with HDM and allergic rhinitis and HDM and allergic asthma [25,42]. Alvarez et al. [42] studied 31 patients with HDM and mild allergic asthma and 15 patients with HDM and allergic rhinitis. The study concluded that patients with HDM and allergic asthma responded to a lower allergen dose and LAR occurred more frequently than patients with HDM and allergic rhinitis. The absolute increase in sputum ECP levels was higher in patients with asthma than in those with allergic rhinitis. Taking these results together, we might speculate that two distinct allergic phenotypes exist: one represented by patients who respond with an EAR only, and one by patients who develop an LAR. However, this has to be proven by future studies.

In recent years, miRNA functions in allergy and asthma have been extensively investigated in order to better understand the molecular mechanisms involved in the development of this heterogeneous disease [43]. In the present study, three distinct miRNAs, miR-15a-5p, miR-15b-5p and miR-374-5p, were analyzed to develop a deeper understanding of the cellular and molecular mechanisms of LAR. As recently described, these miRNAs shared targets like CCND1, VEGFA and GSK3B, which are known to be involved in airway remodeling, basement membrane thickening and extracellular matrix deposition [27]. Therefore, the temporal course of mRNA expression was determined. Interestingly, the expression level of all three miRNAs was significantly downregulated seven hours after BAC.

As mentioned above, the pathophysiology of asthma is due to eosinophilic and Th2 as well as Th1, and Th17 dysregulation and their corresponding cytokines [44]. In this regard, the findings of Wei et al. showed that miR-15a-5p is involved in the regulatory mechanisms of ozone-induced asthma development by promoting airway smooth muscle cell proliferation and Th1/Th2 imbalance [45]. In addition, miR-15a-5p has been proven to be involved in allergic rhinitis immune response by inhibiting IL-13-induced expression of GM-CSF, eotaxin and MUC5AC nasal epithelial cells [46]. The work by Liu et al. further demonstrated that miR-15b-5p inhibits the HMGB1/TLR4/IL-33 signaling pathway and thus acts protectively as against allergic airway inflammation [47]. Moreover, miR-15b-5p inhibits tumor necrosis factor alpha-induced proliferation, migration and extracellular matrix production of airway smooth muscle cells [48].

No or only very weak correlations were found between these miRNAs and the eosinophil count. The reason for this can be explained in part by the small number of measurements and missing results due to shipping problems of eosinophils.

However, all three microRNAs appear to be involved in the modulation of the late allergic response. They regulate inflammatory processes and influence lung function and could represent potential biomarkers or therapeutic targets for the treatment of allergic diseases.

However, this study has some limitations. The number of patients who underwent BAC was small. Still, it may be helpful to mention that all patients were exposed three times to BAC with a day-long follow-up, which implies a high burden for the participants. Including more patients might improve statistics but probably might not change the overall message of this study.

A further criticism might be that the crude percentage of participants exhibiting LAR without a preceding EAR does not provide sufficient evidence of independence. Since the absence of EAR and LAR was based on spirometry (FEV1) by predefined FEV1 cutoffs, this does not exclude an actual inflammatory response in the airways. It is well known that an inflammatory response can be demonstrated after low allergen exposure even in the absence of EAR and LAR [49]. It can be speculated that the definition of EAR and LAR by spirometry is not sufficient, since the inflammatory reaction might be more important than a short drop in FEV1.

In summary, we can confirm that a LAR is induced in a significant portion of patients without a preceding EAR to low allergen exposure. This leads to a strong inflammatory reaction with an increase in eNO. Accordingly, these individuals may be at greater risk of asthmatic symptoms and remodeling with loss of lung function than patients who do not have LAR. To date, these patients can only be detected by BAC. According to the current guideline, BACs are currently only carried out as part of clinical studies at a few specialized centers [25]. In the future, it may be possible to use non-invasive blood sampling (liquid biopsy) to determine distinct RNAs or miRNAs to distinguish allergy sufferers with EAR from those with EAR and LAR [27,50]. Accordingly, the presence of a LAR and surrogate parameters in patients with allergic asthma is an important target for intervention studies such as specific immunotherapy and monoclonal antibodies, especially in patients who are at risk of remodeling with loss of lung function due to LAR.

## 4. Materials and Methods

### 4.1. Study Design

Twenty-five previous patients with asthma and HDM allergy were contacted who were characterized in an earlier AMG study (Eudrac-CT number 2020-000585-41) using bronchial allergen provocation (BAC). The patients suffered from mild allergic asthma that was fully controlled without inhaled corticosteroids (ICS) according to GINA. These patients were informed about the follow-up “investigation of the relationship between the early allergic reaction” (EAR) and the “late allergic reaction” (LAR) in patients with asthma and HDM-allergy after different allergen exposure”. This study was approved after consultation of the Ethics Committee of the Frankfurt Goethe-University (2022-689). After detailed information and written informed consent, 20 of 25 patients agreed to take part in this study. It was a prospective, non-randomized, open study and without a placebo group. Taken together, a BAC was performed on three separate occasions with a tapered allergen dose. The first two challenges were performed during the study (Eudrac-CT number 2020-000585-41); the third BAC was performed approximately 6 months after the first study during the follow-up study (approval 2022-689). Two visits took place between May 2022 and December 2022.

On the first visit, the following investigations were performed: a physical examination, questions about health including the asthma control test (ACT), measurement of eNO and spirometry. Then, blood was drawn for eosinophils, IgE, sIgE and the miRNA expression pattern. This was followed by a BAC with HDM with the individual dose determined in the previous study. Twenty-four hours later, during visit two, eNO, spirometry and eosinophils and miRNA were measured again.

### 4.2. Measurement of Exhaled NO

The measurement of eNO was carried out with the NIOX^®^ (Niox device Circassia AG, Bad Homburg, Germany). The device calculates the eNO value based on the criteria of the American Thoracic Society (ATS) [51].

### 4.3. Pulmonary Function Testing

Spirometry was performed using the Vyntus spirometry device from Vyaire GmbH, Höchberg Germany. The quality standards and norm values of the European Respiratory Society (ERS) were used [52]. The following parameters were determined: forced vital capacity (FVC), forced expiration in 1 s (FEV1) and the Tiffenau Index, the FEV1/VC quotient.

### 4.4. Bronchial Allergen Challenge

The bronchial challenge (BAC) was carried out using the aerosol provocation system (APS) from Vyaire GmbH, Höchberg, Germany, as described in detail by our group [53,54]. To prepare the provocation solution, the lyophilized allergen (708 Dermatophagoides farinae; registration number 467a/87b, batch T7005057-DE) from Allergopharma, Rheinbeck, Germany, was dissolved in 5 mL of physiological saline solution. The final solution contained a concentration of 5000 allergen units (AE)/mL.

Initially, three spirometries were carried out to determine the baseline value (100%) This was followed by a saline inhalation with APS and another spirometry after 2 min. The provocation solution was then nebulized with the allergen dose calculated individually for each patient (single-stage BAC). The APS monitors the inspiratory flow and the inspiration time of the patients using an integrated flow sensor. The system determines the exact amount of aerosol applied per breath. As soon as the specified allergen dose is reached, the APS stops so that only the predefined allergen dose can be administered. After BAC, the spirometry was performed after 10, 15, 30 and 60 min and every hour to assess the EAR and LAR. The maximum drop in FEV1 from 0 to 3 h of EAR was determined for all patients. A decrease of 20% in FEV1 was defined as a significant EAR whereas a LAR was defined as a decrease of FEV1 ≥ 15%, 6–9 h after BAC.

### 4.5. Laboratory Measurements

Blood serum samples (4 mL) were collected and analyzed for eosinophils, total IgE and specific IgE Dermatophagoides pteronyssinus (Dpt) and Dermatophagoides farinae (Dfa) by the ImmunoCAP system (Thermo Fisher Scientific, Dreieich, Germany).

### 4.6. miRNA Next-Generation Sequencing

Blood samples (2 mL) using the PAXgene Blood RNA System for miRNA were collected and stored at −20 °C until analysis. The analysis of miRNAs was performed before, 7 and 24 h after BAC1. The mRNAs (miR-15a-5p, miR-15b-5p, and miR-374a-5p) which showed the highest significance in a previous study were analyzed by qPCR (TaqMan^®^ Advanced miRNA Assays; Thermo Fisher, Dreieich, Germany) as previously described [27]. Briefly, total RNA, including miRNA, was isolated from peripheral blood samples using the Paxgene Blood miRNA Kit (Qiagen, Hilden, Germany) according to the manufacturer’s instructions. The cDNA was prepared by extending 5 ng t-RNA at the 3′ end of the mature transcripts by poly(A) addition. This was followed by extension of the 5′ end by adaptor ligation and universal reverse transcription and amplification using the TaqMan^®^ Advanced miRNA cDNA Synthesis Kit in the thermal cycler (GeneAmp Cycler PCR Systems 9700 v3.12., Thermo Fisher, Dreieich, Germany). The thermocycler program of RT cycles was programmed according to the manufacturer’s description. The miR-24-3p was used as an endogenous control due to its low variance and high abundance compared to the control samples. The qPCR was performed using the StepOnePlus Real-Time PCR System and StepOnePlus™ software v2.3 (Thermo Fisher Scientific, Dreieich, Germany). Analysis was performed using the 2^−ΔΔCt^ method in Expression Suite Software v1.1 (Fisher Scientific, Dreieich, Germany).

### 4.7. Statistics

Statistical data analysis was performed using the SPSS for Windows^®^ 11.0 software package (SPSS Inc., Chicago, IL, USA) and GraphPad PRISM 10.1.2 (GraphPad software, La Jolla, CA, USA). The data are presented as mean + SD and median and range for non-normal distribution. Within individual changes were analyzed using the paired Student *t*-test or corresponding non-parametric test. Changes between patients without EAR and LAR, and patients without EAR but isolated LAR, were analyzed using the unpaired t-test, or corresponding non-parametric tests. Differences between multiple groups were tested using one-way ANOVA or corresponding non-parametric test. For correlation analysis, we used Spearman testing. Statistical significance was defined as *p* < 0.05.

## Figures and Tables

**Figure 1 ijms-26-02088-f001:**
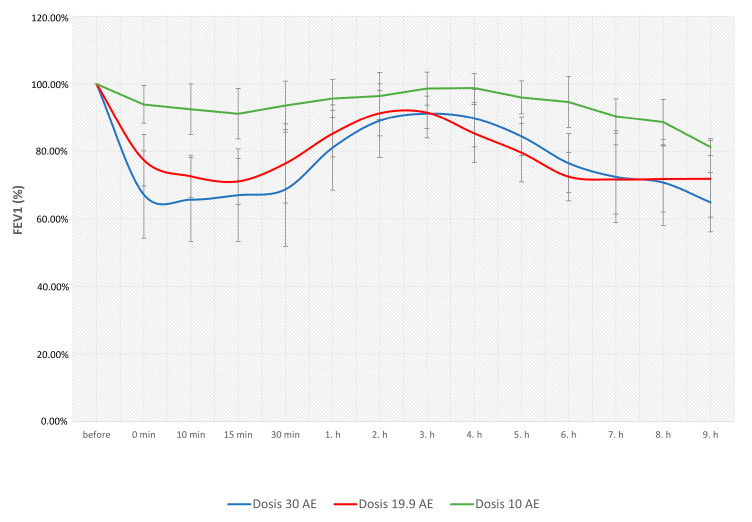
Change of EAR and LAR in response to the allergen reduction.

**Figure 2 ijms-26-02088-f002:**
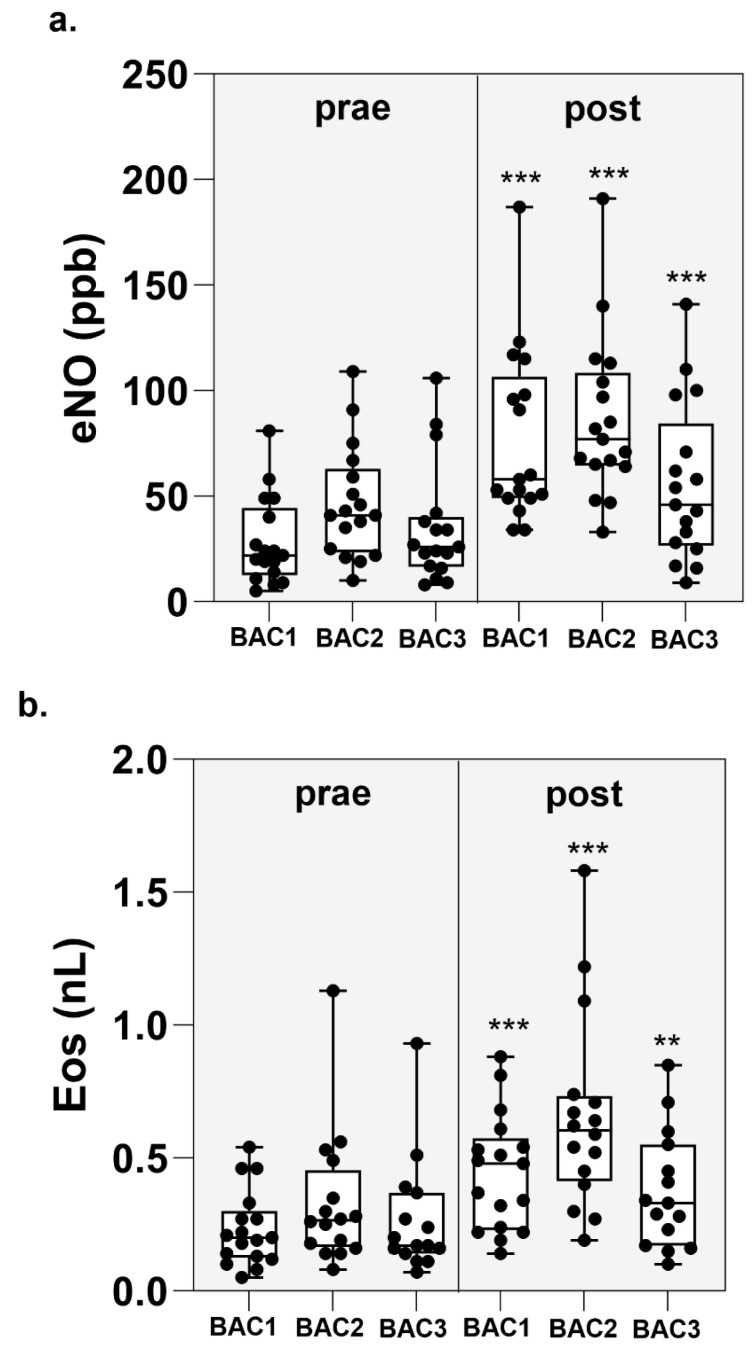
eNo (**a**) and eosinophils (**b**) before and 24 h after BACC. For statistical analysis, BAC pre- was compared to BAC post using the Wilcoxon test. ** *p* < 0.01, *** *p* < 0.001.

**Figure 3 ijms-26-02088-f003:**
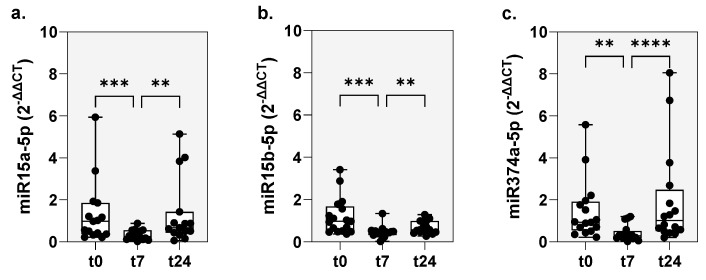
Time course of mRNA expression. Changes in the expression of (**a**) miR-15a-5p, (**b**) miR-15b-5p and (**c**) miR-374a-5p were analyzed in the peripheral blood of patients with asthma and HDM allergy before (t0), 7 (t7) and 24 h (t24) after BAC by RT-PCR. ** *p* < 0.01, *** *p* < 0.001, **** *p* < 0.0001.

**Figure 4 ijms-26-02088-f004:**
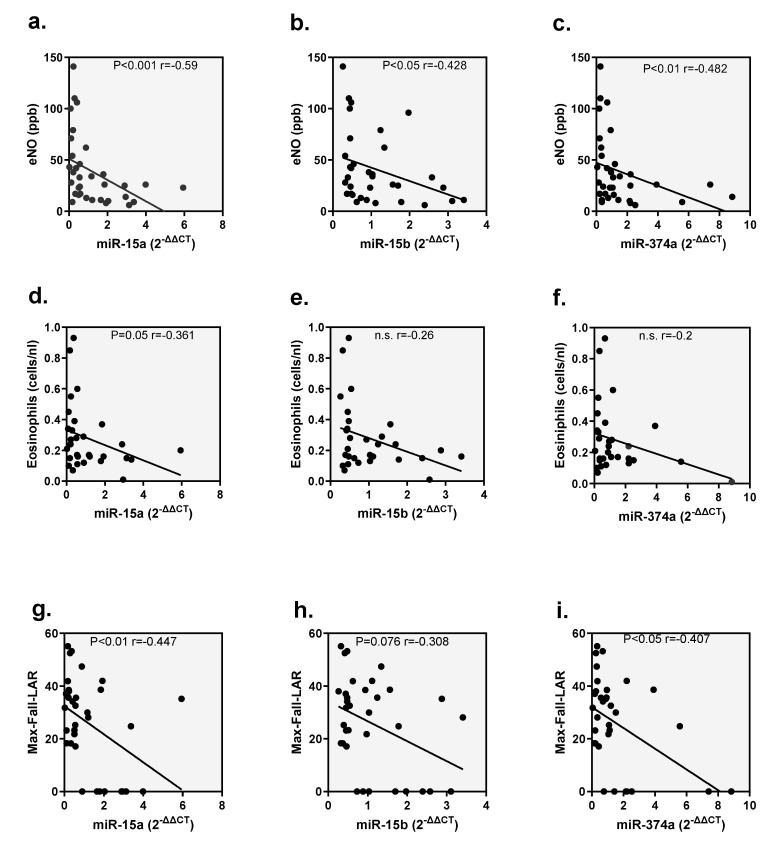
Correlations of miRNA expression with eNO, eosinophilia and max fall LAR. Correlations of miR-15a-5p, miR-15b-5p and miR-374a-5p with (**a**–**c**) eNO, (**d**–**f**) eosinophils and (**g**–**i**) max fall LAR are shown.

**Table 1 ijms-26-02088-t001:** Clinical characteristics of patients with and without LAR.

	Total Patients n = 20	No EAR + LAR n = 10	No EAR, But Isolated LAR n = 7
**Age (years)**	25 (19–33)	24 (19–33)	26 (20–28)
**Gender** (male/female)	9 male 11 female	4 male 6 female	4 male 3 female
**Total IgE** (KU/L)	129 (4–1971)	155.5 (35–1971)	146 (4–532)
**Dfa. sIgE** (KU/L)	15.4 (0.57–78.1)	17.6 (1.68–100)	28.45 (0.57–78.1)
**Dpt. sIgE** (KU/L)	11.8 (1.21–98.8)	11.6 (1.8–98.8)	14.2 (1.21–82.2)
**FVC** (%)	99.3 (79.8–111.9)	104.35 (79.8–111.9)	98.95 (91.4–105.1)
**FEV1** (%)	88.9 (73.9–113.8)	90.95 (73.9–113.8)	83.1 (79.9–103.1)
**PD20** **Methacholin** (mg)	0.22 (0.01–1.72)	0.135 (0.01–1.72)	0.35 (0.01–0.74)

Dermatophagoides farinae (Dfa); Dermatophagoides pteronyssinus (Dpt); specific Ige (sIgE); kilo units per liter; (KU/L); forced vital capacity (FVC); forced expiratory volume in one second (FEV1), methacholine doses required to cause a 20% fall in FEV1 = PD20; medians and range are shown.

**Table 2 ijms-26-02088-t002:** eNO, eosinophils and FEV1 24 h after BAC3.

	No EAR + LAR n = 10	No EAR, But Isolated LAR n = 7	*p*
**Before BAC eNO** (ppb)	10 (8–74)	21 (7–104)	n.s.
**After BAC eNO** (ppb)	43.5 (9–100)	46 (25–110)	n.s.
**Delta eNO** (ppb)	11.5 (1–39)	20 (9–86)	n.s.
**Before BAC Eosinophils** (G/L)	0.17 (0.07–0.93)	0.305 (0.11–0.51)	n.s.
**Alter BAC Eosinophils** (G/L)	0.23 (0.1–0.45)	0.465 (0.21–0.71)	n.s.
**Delta Eosinophile** (G/L)	0.09 (0.01–0.93)	0.22 (0.2–0.3)	*p* < 0.05
**FEV1 before BAC** (%)	90.95 (73.9–113.8)	83.1 (79.9–103.1)	n.s.
**FEV1 24 h after BAC** (%)	87.5 (73.3–109.3)	80.8 (75.1–92)	n.s.
**FEV1 Delta Fall (mL)**	0.06 (0.01–0.48)	0.39 (0.13–0.62)	*p* < 0.01

Early asthmatic reaction (EAR); late asthmatic reaction (LAR); bronchial allergen challenge (BAC); exhaled no (eNO); forced expiratory volume in one second (FEV1), median and ranges are shown. Not significant, n.s.

## Data Availability

The data presented in this study are available on request from the corresponding author. The data are not publicly available due to privacy and ethical concerns.

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
