# Peer review of "The Late Asthmatic Reaction Is in Part Independent from the Early Asthmatic Reactions"

_ijms, 2025, doi:10.3390/ijms26052088_

Round 1

Reviewer 1 Report

Comments and Suggestions for Authors

Thank you for such an interesting study. Studies on exposure to allergens are rare, so this is exceptional work. I was mostly satisfied with the informative yet explanatory writing style. Considering that 20 patients is not a whole study as we all know it, I would suggest adding a pilot study in the text or the title, as I believe this study must be expanded by including more study subjects. 

- "Provocation" I would suggest changing to "challenge" as it is an appropriate word. 

- Table 1 - not sex, but gender and one person is missing out on 20 who are stated to participate in the study. Also, all participants were males? Citing "Sex (male/male)." Under Tables 1 and 2, there is no explanation of D. far. sIgE (KU/L), FVC (%), FEV1, PD20. Please make sure to explain all of the abbreviations. Also, are there any significant differences between IgE and other data? The same applies to Table 2. 

- The miRNA data is challenging to read. Did you consider putting the data into tables or writing results instead of results with ± SEM or SD? 

- The different fonts on the Figures are disturbing. Also, Figure 2 is not informative. Are there any significant differences?

Author Response

ANSWERS TO THE REVIEWERS’ COMMENTS

Dear Editor, Dear Reviewers,

Thank you for your proficient and beneficial comments. We have carefully revised our manuscript in accordance with your suggestions and comments whenever it was possible.

Answers to Reviewer 1

Thank you for such an interesting study. Studies on exposure to allergens are rare, so this is exceptional work. I was mostly satisfied with the informative yet explanatory writing style.

Answer: Thanks for your nice comment.

Considering that 20 patients is not a whole study as we all know it, I would suggest adding a pilot study in the text or the title, as I believe this study must be expanded by including more study subjects. 

Answer: We agree with the reviewer that 20 patients are better classified as a pilot study and mentioned this in the abstract and in the introduction.

- "Provocation" I would suggest changing to "challenge" as it is an appropriate word. 

Answer: We agree with the reviewer that bronchial allergen challenge (BAC) is more often used in literature and recently used in the EAACI position paper (Ioana Agache et al.  Allergy 2022), reference 47.  Therefore, we used the term challenge throughout the text and substituted BAP by BAC.

- Table 1 - not sex, but gender and one person is missing out on 20 who are stated to participate in the study. Also, all participants were males?  Citing "Sex (male/male)." Under Tables 1 and 2, there is no explanation of D. far. sIgE (KU/L), FVC (%), FEV1, PD20. Please make sure to explain all of the abbreviations.

Answer: Sorry for our mistake. We corrected the data and added all abbreviations. Dermatophagoides farinae (Dfa); Dermatophagoides pteronyssinus (Dpt); Specific Ige (sIgE); Kilo Units per liter; (KU/L); Forced vital capacity (FVC); Forced expiratory volume in one second (FEV1), Methacholine doses required to cause a 20% fall in FEV1 = PD20.

Also, are there any significant differences between IgE and other data?

Answer: There were no significant differences, most likely to the small number of patients investigated.

The same applies to Table 2. 

Answer: We added all abbreviations.

- The miRNA data is challenging to read. Did you consider putting the data into tables or writing results instead of results with ± SEM or SD? 

Answer: We have restructured and revised the paragraph and hope that the data is now easier to read. For the presentation of the data, we have chosen the median and range, as not all data is normally distributed

- The different fonts on the Figures are disturbing. Also, Figure 2 is not informative. Are there any significant differences?

Answer: We have completely revised Figure 2 and have added the significances.

Reviewer 2 Report

Comments and Suggestions for Authors

Here I attach the review

Author Response

Reviewer 2

I had to revise the manuscript entitled ~The late asthmatic reaction is in part independent from the 2 early asthmatic reactions, for International Journal of Molecular science.

The study discusses the implication of early and late asthmatic response in HDM asthmatic patients, highlighting the independence of late asthmatic reactions to early ones and their impact on lung functions. It is an interesting article that provides some hypothesis of early and late inflammatory responses after HDM exposure. Some aspect should be revised before the article could be published.

Answer: Thanks for your nice comment.

The abstract is not structured

Answer: The summary has been revised and restructured

The objective of the study is not clearly stated.

Answer: The aim of the study was clearly formulated.

“The aim of the present study was to investigate whether a LAR can occur separately even without significant EAR.”

It is not very clear the protocol 20 patients were provoked with at 3 different time points, or 20 patients were randomly divided in 3 groups that received different doses of HDM at BAP? What was the interval between provocations? In the results section the authors compared the obtained results between BAP1, BAP2 and BAP 3 but they don’t mention what that mean.

Answer: We agree with the reviewer that this is not clear enough explained in the method section. We revised the text as following:

In a pilot study of 20 patients with asthma and HDM allergy, a bronchial allergen challenge (BAC) was performed on three separate occasions with a tapered allergen dose. The first two challenges were performed during the study (Eudrac-CT number 2020-000585-41). Then the patients were invited to take part at a follow-up study “Investigation of the relationship between the early allergic reaction (EAR) and the late allergic reaction"(LAR) in patients with asthma and HDM-allergy after different allergen exposure".  This study was approved after consultation of the Ethics Committee of the Frankfurt Goethe-University (2022-689) and performed approximately 6 months after the first study.

The LAR was assess at 6 or 9 hours?

Answer: After BAC, the spirometry was performed after 10, 15, 30 and 60 minutes and every hour to assess the EAR and LAR. The EAR was defined between 0-3 hours and the LAR between >3-9 hours after BAC. The maximum drop of the FEV1 was recorded for the EAR (0-3 hours after BAC) and for the LAR (<3-9 hours after BAC).

The legend of table 2 is missing (what means * and **).

Answer: Sorry for our mistake, we added a third column to show the significance between the groups and deleted * and **. In addition, we added all abbreviations.

The study highlights the evidence but the limits were not mentioned.

Answer: We agree with the reviewer that the limitations of the study should be mentioned and add a short section of the limitations to the discussion.

Reviewer 3 Report

Comments and Suggestions for Authors

The Article by Zielen et al. submitted for review presents an interesting and current problem of the airways physiology and possibilities of two distinct allergic phenotypes (EAR + LAR vs LAR) in patients with asthma following bronchoprovocation with specific (house dust mite extract) factors. It should be noted that this topic is new and there is no data in the available literature examining this issue. This problem seems to be also important and undervalued in the period where the “classical” and “biological” treatment of asthma is widely used. The article is written concisely, and despite the extensive experimental part (e.g. lot of epidemiological data, laboratory data, spirometric results) easy to read and understand, it does not contain unnecessary repetitions and ambiguities. However, there are minor comments, that I consider important to change in the manuscript:

1. Materials and methods section – line 100 – on the basis of which criteria, asthma was diagnosed (GINA?).

2. Laboratory measurements – line 145 – what was the volume of blood collected and into which tubes was collected.

3. miRNA next generation sequencing – line 153 – is it possible to determine which specific cells were used to perform miRNA analysis

4. Discussion sectionlines 287, 296 – expand the abbreviations (ECP, CCND1, VEGFA, GSK3B).

5. In the Discussion section – is worth to mention some cells and cytokines that are important in asthma – https://doi.org/10.3390/cells12091326

6. You also do need to check the paper for typing errors, punctuation errors, use of capital letters, spacing, units etc.

7. In the summary, it is also worth to mention the limitations and emphasize the strengths of the article.

Congratulations on your excellent work, please include the above comments in the revised version.

Best regards.

Author Response

Reviewer 3

The Article by Zielen et al. submitted for review presents an interesting and current problem of the airways physiology and possibilities of two distinct allergic phenotypes (EAR + LAR vs LAR) in patients with asthma following bronchoprovocation with specific (house dust mite extract) factors. It should be noted that this topic is new and there is no data in the available literature examining this issue. This problem seems to be also important and undervalued in the period where the “classical” and “biological” treatment of asthma is widely used. The article is written concisely, and despite the extensive experimental part (e.g. lot of epidemiological data, laboratory data, spirometric results) easy to read and understand, it does not contain unnecessary repetitions and ambiguities. However, there are minor comments, that I consider important to change in the manuscript:

Answer: Thanks for your positive judgement.

  1. Materials and methods section – line 100 – on the basis of which criteria, asthma was diagnosed (GINA?).

Answer: Patients suffered from mild allergic asthma according to GINA. We added the following explanation to the methods section: Patients suffered from mild allergic asthma and were fully controlled without inhaled corticosteroids (ICS) according to GINA.

  1. Laboratory measurements – line 145 – what was the volume of blood collected and into which tubes was collected.

Answer: Laboratory measurements: Blood serum samples (4 ml) were collected and analysed for ….

miRNA Next generation sequencing: Blood samples (2 ml) using the PAXgene Blood RNA System for miRNA were collected and stored at -20°C until analysis.s.

  1. miRNA next generation sequencing – line 153 – is it possible to determine which specific cells were used to perform miRNA analysis

Answer: For the miRNA analysis, whole blood was collected in PAXgene tubes and miRNA was isolated from the entire blood cell repertoire. It is therefore difficult to draw conclusions about specific cell populations.

  1. Discussion section – lines 287, 296 – expand the abbreviations (ECP, CCND1, VEGFA, GSK3B).

Answer: We added ECP, Eosinophilic Cationic Protein; CCND1, Cyclin D1; VEGFA, Vascular endothelial growth factor A; GSK3B, Glykogensynthase-Kinase 3 to the abbreviation list.

  1. In the Discussion section – is worth to mention some cells and cytokines that are important in asthma – https://doi.org/10.3390/cells12091326

Answer: We thank the reviewer for the suggestion and have once again emphasized the important immune cells and discussed them with the miRNA findings.

  1. You also do need to check the paper for typing errors, punctuation errors, use of capital letters, spacing, units etc.

Answer: We checked the paper for typing errors, punctuation errors by a native speaker.

  1. In the summary, it is also worth to mention the limitations and emphasize the strengths of the article.

Answer: We agree with the reviewer that the limitations of the study should be mentioned and add a short section of the limitations to the discussion.

Congratulations on your excellent work, please include the above comments in the revised version.

Reviewer 4 Report

Comments and Suggestions for Authors

The study conducted bronchial allergen provocation three times with a tapered allergen dose and measured bronchial inflammation markers including eNO, eosinophils and miRNAs, both before, and 24 hours after BAP1. The authors employed relatively simple methods to test the unadjusted changes in FEV1 across BAP1-3, as well as changes in eNO and eosinophils before and 24 hours after BAP. Additionally, they examined changes in mRNA expressions over different time points; correlations between the inflammation markers were also tested. 
I have several major concerns about the study, particularly concerning its objectives and the interpretation of the results. One primary issue is the small sample size and the statistical analyses, which lack adjustment for potential covariates. Furthermore, the research gap and clinical and public health implications were not sufficiently or clearly articulated in either the abstract or the background section. The abstract also presents unclear descriptions of the results and needs to be further revised. Last, several statistical methodological details require clarification.

1.    Line 27-31: Revision is needed for clarity. Specifically: 
a.    Please clarify what “this” refers to in line 27
b.    Define “these individuals” in line 29 to avoid ambiguity

2.    Line 35-36: The statement “the current knowledge believes that the EAR always precedes the LAR” is not entirely accurate. Studies from over 30 years ago have documented isolated late responses. For example, Durham (1990, Late asthmatic responses. Respiratory medicine, 84(4), 263-268.), stated: 
“Late responses occurring in the absence of a preceding early response have been observed following occupational type challenge tests for many years. However,
recent information suggests that these isolated late responses may also occur frequently following exposure to common allergens at low concentrations, insufficient to provoke an immediate asthmatic response”.
Please revise this statement accordingly.

3.    Line 40-41: 
a.    Please clarify which lung function measure was used, was it FEV1?
b.    What are the meanings of numbers “13.37” and “525”? Are they standard deviations? Also, “525” appears to be a typo, and “+” should likely be “+/-”.
c.    Specify whether the reported decrease is statistically significant and indicate the significance level.

4.    Line 43: The explanation of how the LAR was independent of the EAR is unclear. Please provide further clarification.

5.    Line 44-48: Given the small sample size (N = 20), the crude percentage of participants exhibiting LAR without a preceding EAR does not provide sufficient evidence of independence. A clearer and more structured presentation of the results in the abstract is needed to highlight the study’s key findings.

6.    Line 93-94: As noted earlier, the authors should provide a stronger justification for the research gap and the public health/clinical implications of the independence between EAR and LAR.

7.    Line 166-174:
a.    Please explain why the mean is presented as mean + SD rather than mean +/- SD?
b.    For non-normally distributed variables, specify the reported range and provide a justification. Does the range represent the minimum–maximum values, the interquartile range (25th–75th percentiles), or another measure?
c.    Did "groups" refer to BAP at different doses, different time points before and after BAP, or both (depending on the specific test)? Please clarify the definition of "group" in both within-group and between-group analyses.

Author Response

Review 4

The study conducted bronchial allergen provocation three times with a tapered allergen dose and measured bronchial inflammation markers including eNO, eosinophils and miRNAs, both before, and 24 hours after BAP1. The authors employed relatively simple methods to test the unadjusted changes in FEV1 across BAP1-3, as well as changes in eNO and eosinophils before and 24 hours after BAP. Additionally, they examined changes in mRNA expressions over different time points; correlations between the inflammation markers were also tested. 

I have several major concerns about the study, particularly concerning its objectives and the interpretation of the results. One primary issue is the small sample size and the statistical analyses, which lack adjustment for potential covariates.

Furthermore, the research gap and clinical and public health implications were not sufficiently or clearly articulated in either the abstract or the background section.

The abstract also presents unclear descriptions of the results and needs to be further revised. Last, several statistical methodological details require clarification.

  1.    Line 27-31: Revision is needed for clarity. Specifically: 
    a.    Please clarify what “this” refers to in line 27
    b.    Define “these individuals” in line 29 to avoid ambiguity

  2.    Line 35-36: The statement “the current knowledge believes that the EAR always precedes the LAR” is not entirely accurate. Studies from over 30 years ago have documented isolated late responses. For example, Durham (1990, Late asthmatic responses. Respiratory medicine, 84(4), 263-268.), stated: 
    “Late responses occurring in the absence of a preceding early response have been observed following occupational type challenge tests for many years. However,
    recent information suggests that these isolated late responses may also occur frequently following exposure to common allergens at low concentrations, insufficient to provoke an immediate asthmatic response”.
    Please revise this statement accordingly.

Answer: We appreciate the comment of the reviewer and revised this section according to your advice.

  1.    Line 40-41: 
    a.    Please clarify which lung function measure was used, was it FEV1?

Answer: Yes, we used spirometry and recorded FEV1.

  1.    What are the meanings of numbers “13.37” and “525”? Are they standard deviations? Also, “525” appears to be a typo, and “+” should likely be “+/-”.

Answer: This was corrected

  1.    Specify whether the reported decrease is statistically significant and indicate the significance level.

Answer: We added the significance to the Figures.

  1.    Line 43: The explanation of how the LAR was independent of the EAR is unclear. Please provide further clarification.

Answer: We added the following to the introduction: If two distinct phenotypes of HDM allergy are present, this may have important clinical and public health implications. The phenotype with EAR only, may be less likely to develop chronic asthma whereas the LAR phenotype is at high risk to progress to chronic asthma with lung function loss.

and the following to the discussion: Moreover, a silent trigger below the threshold of an EAR is barely noticeable and avoidable for the affected person after low allergen exposure. This phenotype constellation, that in some patients with allergic asthma a LAR is triggered even with low allergen exposure, certainly may contribute to the chronicity and remodelling of the airways with loss of lung function.

However, this must be proven by further studies that two distinct phenotypes of HDM allergy are present. One, where the LAR is triggered by high and low allergen exposure and one where no LAR can be induced. This phenotype is less likely to develop chronic asthma. 

  1.    
    Line 44-48: Given the small sample size (N = 20), the crude percentage of participants exhibiting LAR without a preceding EAR does not provide sufficient evidence of independence.

Answer: We agree with the reviewer that the sample size was small and that the high number of patients exhibiting LAR without a preceding EAR does not provide sufficient evidence of independence. Absence of EAR and LAR was based on spirometry (FEV1) does not exclude an actual inflammatory response in the airways.

However, all patients were exposed three times to allergen challenge and daylong follow-up which implies a burden for the participants. Including more patients might improve statistics but probably will not change the overall message of this study. Nevertheless, we discussed both points in the new section of limitation of the study.   

A clearer and more structured presentation of the results in the abstract is needed to highlight the study’s key findings.

Answer: We agree with the reviewer and the abstract was revised.

  1.    Line 93-94: As noted earlier, the authors should provide a stronger justification for the research gap and the public health/clinical implications of the independence between EAR and LAR.

Answer: As mentioned above we revised the paper in the introduction and discussion to highlight that two distinct allergic phenotypes may exist: one who respond with an EAR only and one who develops an LAR. However, as mentioned by you and reviewer 1 our study should be considered as a pilot study. Further studies are needed to prove that two distinct allergic phenotypes are present, one with EAR and one with EAR plus LAR, which is activated by low allergen exposure and causing a high inflammatory reaction.

This discussion was considered in the section limitations of the present study.

  1.    Line 166-174:

  2.    Please explain why the mean is presented as mean + SD rather than mean +/- SD?

Answer: Changed to +

  1.    For non-normally distributed variables, specify the reported range and provide a justification. Does the range represent the minimum–maximum values, the interquartile range (25th–75th percentiles), or another measure?

Answer: The range, describing the minimum and maximum value, is a well-established term. I agree that if any other measure were used, a specification would be necessary.

  1.    Did "groups" refer to BAP at different doses, different time points before and after BAP, or both (depending on the specific test)? Please clarify the definition of "group" in both within-group and between-group analyses.

Answer: We replaced “within-group” with “within-individual” and clarified that between-group comparison is applied when testing differences between patients without EAR and LAR and patients without EAR, but isolated LAR.

Round 2

Reviewer 2 Report

Comments and Suggestions for Authors

The authors significantly improved the article. It should be published in the present form after it is included in the template